# Pan-Genome Analysis Reveals Host-Specific Functional Divergences in *Burkholderia gladioli*

**DOI:** 10.3390/microorganisms9061123

**Published:** 2021-05-22

**Authors:** Hyun-Hee Lee, Jungwook Park, Hyejung Jung, Young-Su Seo

**Affiliations:** 1Department of Integrated Biological Science, Pusan National University, Busan 46241, Korea; ehyuna92@pusan.ac.kr (H.-H.L.); jjuwoogi@pusan.ac.kr (J.P.); jhj4059@pusan.ac.kr (H.J.); 2Environmental Microbiology Research Team, Nakdonggang National Institute of Biological Resources (NNIBR), Sangju 37242, Korea

**Keywords:** *Burkholderia gladioli*, comparative genomics, pan-genome, plant pathogen, human pathogen

## Abstract

*Burkholderia gladioli* has high versatility and adaptability to various ecological niches. Here, we constructed a pan-genome using 14 genome sequences of *B. gladioli*, which originate from different niches, including gladiolus, rice, humans, and nature. Functional roles of core and niche-associated genomes were investigated by pathway enrichment analyses. Consequently, we inferred the uniquely important role of niche-associated genomes in (1) selenium availability during competition with gladiolus host; (2) aromatic compound degradation in seed-borne and crude oil-accumulated environments, and (3) stress-induced DNA repair system/recombination in the cystic fibrosis-niche. We also identified the conservation of the rhizomide biosynthetic gene cluster in all the *B. gladioli* strains and the concentrated distribution of this cluster in human isolates. It was confirmed the absence of complete CRISPR/Cas system in both plant and human pathogenic *B. gladioli* and the presence of the system in *B. gladioli* living in nature, possibly reflecting the inverse relationship between CRISPR/Cas system and virulence.

## 1. Introduction

The impact of environmental variability is an important issue in the evolution of microbial genomes, contributing to phylogenetic position, as well as the diversity of metabolic capabilities to which it is adapted [1,2]. Environmental pressure for genomic evolution can include pH, temperature, oxygen, nutrient availability, competition with other bacteria, and stress-inducing defense mechanisms, from habitats and hosts [2,3,4]. The changes in the microbial genome are related to several evolutionary events: (1) the loss of genes by deletion, (2) the modification of gene products by mutation, and (3) the acquisition of new genes by lateral gene transfer and duplication [5]. Functional diversity through genomic evolution can either confer on bacteria a unique survival strategy in harsh environments or the establishment of pathogenicity to dominate the hosts [3]. Hence, the genomic architecture of microbes shows signatures of a long journey of adaptive evolution for different specialized lifestyles. Even in the taxonomically distant lineage, microbes within similar lifestyles exhibit similar genomic inventory to acclimatize to common environmental conditions in each niche [3,6]. Nevertheless, information on bacterial genomic diversity regarding dynamic interactions between the microbe and habitat or host conditions is currently limited.

Rapid advances in next-generation sequencing (NGS) technology have drastically reduced the cost of genome sequencing, thereby accelerating the development of comparative genomics through numerous genomes of various organisms [7]. In contrast to traditional analyses focusing on a single genome, comparisons of whole genomes provide plentiful genetic information to elucidate genomic structural landmarks, novel gene repertoires, and phylogenetic relationships among different organisms [8,9]. The fundamental goal of comparative genomics is to build a pan-genome map based on the genomes, as first quoted by Tettelin et al. in 2005 [10]. The pan-genome has a conserved core genome shared among approximately all members of the target group, and an accessory and dispensable genome, which are either unique to individual organisms or present in variable regions among two or more but not all genomes [11,12]. While the core genome includes genes essential for its basic lifestyle and major phenotypic traits in all targets, those genes appearing in the accessory and dispensable genome are involved in the strain-specific phenotypes such as adaptability, pathogenicity, and stress responses for survival in particular niches [11]. In some studies, such comparative genomic analysis is effective in tracking the novel niche-specific adaptive strategy of microbes of interest [13,14,15,16].

The genus *Burkholderia*, Gram-negative, aerobic, motile, and rod-shaped bacteria, includes members that are characterized by high versatility and adaptability to various ecological niches [17,18]. This ecological versatility within and between species of *Burkholderia* is due to their genome plasticity, unusually large multiple genomes, and potential for intragenomic rearrangement between chromosomal replicons [19]. *B. gladioli* are also widely recognized as a bacterium that exhibits remarkable divergence of ecological niches even within species [20], and the first whole-genome sequence was reported in rice pathogenic *B. gladioli* BSR3 in 2011 [21]. Although *B. gladioli* was initially known as a plant pathogen of gladiolus-causing rot diseases [22], currently, *B. gladioli* is better known as the rice pathogen that consistently has threatened rice farming in Japan, China, and South Korea [23,24], inducing symptoms of seedling blight, panicle blight, and leaf-sheath browning in rice [25]. Moreover, *B. gladioli* are isolated not only in other plants (onions, iris, and mushrooms) [26,27] but are also found in diverse habitats, including soil, environmental water, [28,29], and even the respiratory tract of immunosuppressed humans [30,31]. 

The current study aimed to understand the focused biological pathways that were previously unknown but could be related to the successful adaptation of *B. gladioli* in each niche via functional pan-genomic analyses. For an unbiased investigation of the pan-genome’s biological function, we carefully selected and compared 14 *B. gladioli* genomes that can represent different lifestyles within the gladiolus, rice, humans, and natural environment. A phylogenomic analysis of *B. gladioli* strains was firstly conducted, showing the close relationship between the *B. gladioli* genome and ecological niche. Based on this relationship, we constructed the *B. gladioli* pan-genome consisting of core, accessory, dispensable, and unique genomes. Finally, the core and niche-associated genome of the *B. gladioli* pan-genome revealed core biological features regardless of the niche, and adaptive features reflecting habitat ubiquity of *B. gladioli*, respectively.

## 2. Materials and Methods

### 2.1. Public Genomic Resources

For comparative analysis, the genomic sequences for *B. gladioli* strains, including a *B. gladioli* KACC 11889 whole genome sequence, which was previously developed [16], were downloaded from the FTP site of Reference Sequence (RefSeq) database at NCBI (ftp://ftp.ncbi.nih.gov/genomes/, accessed on 10 February 2021). Subsequently, in selecting genomes for the comparative analyses, we excluded as many as possible bacterial genomes which were isolated from unclear or unshared sites with other strains and were previously studied [32]. Consequently, fourteen *B. gladioli* genomes, which were consisted of 4 complete genomes and 10 draft genomes, were carefully included in this study. 

### 2.2. Bacterial Strain, Culture Conditions, and Genomic DNA Extraction

*B. gladioli* KACC 18962 was streaked on plates of Luria-Bertani (LB) agar and incubated at 30 °C for 2 days. After confirmation of pure culture, single colonies were transferred to fresh LB broth and incubated in a shaking incubator at 200 rpm and 30 °C.

Genomic DNA was extracted from *B. gladioli* KACC 18962 using a Wizard Genomic DNA purification kit (Promega, Madison, WI, USA) according to the manufacturer’s recommended protocol. The quantity and purity were determined using a NanoDrop™ 2000 spectrophotometer (Thermo-Fisher Scientific, Waltham, MA, USA). 

### 2.3. Genome Sequencing and Assembly

The whole genome sequencing library was prepared with a SMRTbell template prep kit (Pacific Biosciences, Menlo Park, CA, USA), and then, single-molecule real-time (SMRT) sequencing was conducted on a PacBio RS II platform (Pacific Biosciences) using Macrogen (Seoul, South Korea). To construct more accurate contigs, HiSeq3000 (Illumina, San Diego, CA, USA) paired-end reads were applied for sequence compensation. *De novo* assembly with the hierarchical genome assembly process (HGAP) version 3 was implemented in the PacBio SMRT Analysis algorithm version 2.3.0 package [33]. Quality control of long reads was performed using a PreAssembler filter version 1 protocol from HGAP, and further quality improvements of genome sequences were achieved by polishing with Quiver [33]. To reinforce the genome assembly, HiSeq reads were mapped twice against the PacBio assembly using Pilon version 1.21 [34]. Gene prediction and annotation were initially performed by Prokka version 1.13 [35]. Next, the genome sequence was re-annotated with the NCBI prokaryotic genome annotation pipeline based on the best-placed reference protein set and GeneMarkS+ [36] and finally deposited in the NCBI RefSeq database [37].

### 2.4. Phylogenomic Analysis

For the phylogenomic analysis, whole-genome sequences of *B. gladioli* strains were compared by computing the average nucleotide identity (ANI) and the corresponding alignment coverage using the Python3 module pyani (version 0.2.10; option ‘-m ANIb’) [38]. The ANI values were visualized by heatmap.2 function in the gplots package in the R environment (https://CRAN.R-project.org/package=gplots, accessed on 10 February 2021). We also confirmed the phylogenetic distribution of the 14 *B. gladioli* in cladistic structure, which were previously well described by Jones et al. [32]. To cover the full diversity of the population, the representative 22 *B. gladioli* members were selected from different clades of 206 *B. gladioli* isolates and compared with 14 *B. gladioli* as in a study of Lin et al. [39]. The GenBank annotations were firstly converted to GFF3 format using BioPerl script (bp_genbank2gff3.pl). The GFF3 files were used as input for the Roary version 3.11.2 [40] to extract core genes and to align concatenated core genes with MAFFT using default parameters. In turn, core gene-based phylogenetic trees were generated using RAxML version 8.2.10 [41] with GTR + GAMMA model and bootstrap values from 1000 replications. The phylogenetic tree was visualized by MEGA X [42].

### 2.5. Pan-Genomic Analysis

All the orthologous pairs across 14 *B. gladioli* genomes were calculated using the pan-genome analysis pipeline (PGAP) version 1.2.1 [43] to identify pan-genomes, including the core, accessory, dispensable and unique genome. Under the GF method, the total protein sequences of each strain were mixed together and marked as the strain identifiers. BLAST searches were performed using the BLASTALL program [44] among the mixed protein sequences, and the minimum score value and E value applied in BLAST were set to 40 and 1.0 × 10^−5^, respectively. The filtered BLAST results were clustered by the Markov Cluster algorithm [45], which has been widely used in other studies on prokaryotic genomes to search for orthologs among multiple strains. For grouping the same genes into the same cluster, the global match regions have a minimum of 50% coverage and 50% identity on the protein sequences. The COG distribution in pan-genome was analyzed with the parameter “-function.” The pan-genome characteristic curves were drawn using PanGP version 1.0.1 [46]. Validation of the *B. gladioli* pan-genome was performed by anvi’o platform version 6.2 following the tutorial for microbial pan-genomics [47].

### 2.6. Functional Enrichment Analyses 

Functional annotations of the *B. gladioli* pan-genome were performed using three databases: Clusters of orthologous groups (COG) [48], gene ontology (GO) [49], and Kyoto encyclopedia of genes and genomes (KEGG) [50]. For the annotation of 14 *B. gladioli* genes with COG, a whole genome BLASTP search was performed against a local version of the NCBI COG database (ftp://ftp.ncbi.nih.gov/pub/COG/COG2014/data, accessed on 10 February 2021) with the following parameters: E value < 1.0 × 10^−5^; identity more than 30%; coverage more than 30%. Only one annotation per protein was ranked by the E value, identity, and coverage. GO annotation was performed using a java-based Blast2GO software version 5.2.5 [51] with the default parameters. Blast2GO annotation was initiated by BLAST searches against the NCBI RefSeq non-redundant protein database followed by Interproscan [52] analyses to identify conserved protein domains. Finally, the GO IDs of all proteins were retrieved from the Blast2GO annotation database. This functional annotation categorized the *B. gladioli* proteins under all GO categories that include cellular components, molecular processes, and biological processes. The protein of each genome was also functionally annotated using pre-computed hidden Markov model profiles of KEGG ortholog by kofamscan version 1.2.0, which assigns KEGG orthology identifier to proteins via HMMER/HMMSEARCH. The kofamscan output in a mapper format was processed by the KEGG Mapper-Reconstruct Pathway utility with manual curation.

After COG/GO/KEGG functional annotation, enrichment analyses for each database were subsequently performed to find significantly over-represented biological annotations in genomes. The pan-, core and niche-associated genomes of *B. gladioli* were subjected to the COG, GO, and KEGG pathway enrichment analyses, respectively. All enrichment analyses were performed using a hypergeometric test with the proper function contained in the R statistical environment [53]. Specifically, the hypergeometric test considered the following statistics: the number of genes involved in the COG/GO/KEGG systems, the percentage of the systems covered by a specific category, and the expected number of target genes in the specific categories. The COGs, GO terms, and KEGG pathways with corrected *p*-values less than 0.05 were considered significantly enriched.

### 2.7. Identification of Secondary Metabolite Biosynthetic Gene Clusters and CRISPR/Cas

Screening for secondary metabolite biosynthetic gene cluster (BGC) was performed on the antibiotics and secondary metabolite analysis shell (anti-SMASH, https://antismash.secondarymetabolites.org/, accessed on 10 February 2021) of the online web server [54]. For the screening, individual *B. gladioli* genome sequences were submitted to the anti-SMASH server in the GenBank.

The clustered regularly interspaced short palindromic repeat (CRISPR) regions and CRISPR-associated (Cas) proteins were identified by CRISPRCasFinder (https://crisprcas.i2bc.paris-saclay.fr/CrisprCasFinder, accessed on 10 February 2021) [55]. The whole genome sequences of *B. gladioli* strains were submitted to the CRISPRCasFinder tool in fasta format. For CRISPR array identification, 100 bp was selected as the size threshold of the flanking region, and the CRISPR array contained truncated repeats. Only those with an evidence level ≥1 were considered in this study. The spacer sequences, which were detected from CRISPRCasFinder, were subsequently entered into the CRISPRTarget tool [56] to predict the most likely targets of CRISPR RNA. A cut-off score of 30 was applied and we manually examined the results to exclude self-matching or accidental matching results.

## 3. Results and Discussion

*B. gladioli* are unique pathogenic bacteria that can cause diseases in both plants and humans, and they occupy significantly divergent ecological niches. Specifically, even strains of the same *B. gladioli* species exhibit these unique characteristics. Recent comparative genomic studies have demonstrated that bacterial adaptation to the environment in the host is closely related to biological capabilities, which are changed by the result of gene gain/loss or genome reduction/expansion [1,5]. Thus, the fundamental aims of this study are to identify concentrated biological functions of the *B. gladioli*, originating from different sources, and to infer the role of these functions, possibly contributing to successful adaptation in the surroundings.

To achieve this goal, we employed the pan-genomic analysis using a total of 14 *B. gladioli* isolates which have been historically, phenotypically, or genomically well characterized. Particularly, of fourteen *B. gladioli*, ten pathogenic *B. gladioli* isolates were classified by niches (gladiolus, rice, and human), and the others were integrated into the nature group. The gladiolus isolates included the pathovar reference strains of *B. gladioli* pv. *gladioli* (ATCC 10248 and NCTC 12378) [57,58,59], *B. gladioli* KACC 11889 [60] and *B. gladioli* ATCC 25417 [61,62] in which several virulence properties were experimentally investigated. Rice isolates comprised *B. gladioli* BSR3, one of the most notable rice pathogenic *Burkholderia* species [21,63,64,65], and *B. gladioli* KACC 18962, which was isolated in this study, presenting similar phenotype traits with strain BSR3. Human isolates were composed of opportunistic human pathogenic and multidrug-resistant *B. gladioli* (AU0032, AU26456, AU29541 and AU30473) in which antibiotic susceptibility profiling has been recently undertaken [66,67]. In addition, *B. gladioli*, which were isolated from water and soil for industrial application [28], were included in nature isolates.

### 3.1. Available Genomic Information for B. gladioli

We sequenced the genome of *B. gladioli* KACC 18962, which was isolated in rice cultured in South Korea using PacBio RS II sequencing technology. Table 1 summarizes several key features for the fully sequenced *B. gladioli* KACC 18962 genome and Figure 1 shows the circular replicons of strain KACC 18962. The complete genome of strain KACC 18962 consists of two chromosomes and one plasmid. The average coverage of PacBio reads was 190-fold for chromosome 1, 187-fold for the chromosome 2, and 190-fold for the plasmid with 92.0% and 83.0% of the reads exhibiting an average Phred quality score ≥Q20 and Q30, respectively (Table 1). A total of 7475 genes were identified from the genome, 173 of which were pseudogenes. The three replicons encode 7216 coding genes, 5 rRNA operons, and 67 tRNA loci. The sequences of *B. gladioli* KACC 18962 chromosome 1, chromosome 2, and plasmid genome have been deposited in GenBank under accession numbers CP045573, CP045574, and CP045575, respectively.

Finally, we collected the genome sequences of 14 *B. gladioli* strains and listed detailed information of them in Table 2. According to the NCBI database, the *B. gladioli* strains were isolated from gladiolus plant (ATCC 10248, KACC 11889, ATCC 25417 and NCTC 12378), rice plant (BSR3 and KACC 18962), human sputum (AU0032, AU26456, AU29541, and AU30473), and in nature (Coa14, MSMB1756, FDAARGOS_390, and FDAARGOS_391) (hereinafter “gladiolus isolates”, “rice isolates”, “human isolates”, and “nature isolates”, respectively). Of these, four strains, ATCC 10248, KACC 11889, BSR3 and KACC 18962, were fully sequenced. Genome sizes ranged between 8.4 Mb (strain NCTC 12378) and 9.3 Mb (strain ATCC 25417) for gladiolus isolates, 8.6 Mb (strain KACC 18962) and 9.1 Mb (strain BSR3) for rice isolates, 8.0 Mb (strain AU0032) and 8.4 Mb (strains AU29541) for human isolates, and 8.2 Mb (strain MSMB1756) and 8.8 Mb (strain FDAARGOS_390) for nature isolates. This genome size variation shows that gladiolus isolates have the largest variation in genome size with 0.9 Mb and human isolates have the smallest genome size variation with 0.4 Mb than the strains isolated elsewhere. This tendency was also observed in GC content and the counts of protein coding genes of gladiolus and human isolates. GC content variations were 67.3–68.0% (gladiolus isolates) and 68.2–68.3% (humans isolates). The number of protein coding genes for gladiolus isolates ranged between 7098 (strain NCTC 12378) and 7941 (strain ATCC 25417), and that for human isolates ranged between 7112 (strain AU0032) and 6769 (strain AU29541). This result indicated that there was no sharp distinction between *B. gladioli* strains in the genomic characteristics (i.e., genome size, coding density and GC content), and there was no clear association between the genomic architecture and isolation source. Therefore, we needed to inspect genome sequences in detail to find clues of niche-directed evolution in genomic contents. 

### 3.2. Phylogenomic Analysis

To evaluate the adequacy of studying the relationship between *B. gladioli*’s genome and niche, a whole genome-based phylogenomic tree was constructed from 14 *B. gladioli* genomes via the average nucleotide identity ANI approach [69], the robust measurement of genomic relatedness between strains (Figure 2A and Appendix A). The most notable result of this phylogenomic tree was highly close relationships of gladiolus isolates (Figure 2A). This result implies that niche could influence bacteria genome shaping especially in gladiolus-originated *B. gladioli*. Meanwhile, gladiolus and human isolates also relatively closed to each other, although pairwise ANI values across gladiolus isolates were more than 99.9%, and those between human isolates were over 98.9% (Appendix A). ANI values between strain AU30473 of human isolates and gladiolus isolates (more than 99.1%) were slightly higher than those between strain AU30473 and other human isolates (over 98.9%). The close relationship between human isolates and gladiolus isolates offers a perspective on the origin and acquisition of biological functions that will be necessary for survival or adaptation in a new challenging niche like humans. According to this perspective, studies by Jones et al. [32] and Lipuma et al. [70], suggested that the main source of human-associated *Burkholderia* is the natural environment. In rice isolates, *B. gladioli* BSR3 and KACC 18962 were clearly distinguished from strains in gladiolus and human isolates while KACC 18962 shared high ANI scores with FDAARGOS_390 and FDAARGOS_391 of nature isolates (ANI values >99.0%). The nature isolates (MSMB1756, Coa14, FDAARGOS_390, and FDAARGOS_391) exhibited limited genome relatedness among each other, presenting ANI values ranging from 97.0–99.0%. 

The ANI-based phylogenomic relationships also validated in cladistic structure, which were previously well described by Jones et al. [32]. We obtained 4392 concatenated core genes by comparing 14 *B. gladioli* genomes in this study with the genomes of 22 previously reported (Figure 2B). As a result, both gladiolus and human isolates were positioned in clade 3. In clade 3, gladiolus isolates still revealed notably close relationship with BCC0771 (*B. gladioli* pv. *gladioli*). However, rice and nature isolates failed again to be grouped into same clade, respectively. The rice isolate BSR3 and nature isolate MSMB1756 were solely belonged to clade 1B and 1A, respectively. The rice isolate KACC 18962 was classified into clade 2 with FDAARGOS_390, and FDAARGOS_391 of nature isolates. The rest 22 *B. gladioli* isolates were positioned identically as the previous classification [39]. In conclusion, the results highlight potential high relatedness between isolation sites and genome contents, although it is difficult to apply this association to all *B. gladioli* isolates.

### 3.3. Pan-Genome Analysis

We performed comprehensive pan-genome analysis for defining the sets of core, accessory, dispensable, and unique genomes. The core genome was defined as the set of genes, shared by all 14 *B. gladioli* strains. The accessory genome, which was available in one or up to n-1 of the genomes (n: total genomes), was composed of the dispensable genome (genes present in two or more strains) and the unique genome that contains the set of genes, observed in only one genome. First, cumulative curves, which were generated by PanGP [46] according to Heaps’ law, were presented in Figure 3A,B. Tettelin et al. [10] reported that a finite or infinite pan-genome can be determined by a prediction using Heaps’ law. Heaps’ law is devised as n = κ*N*^γ^, where *n* is the pan-genome size, *N* is the number of genomes used, and κ and γ are the fitting parameters. For γ < 0, the pan-genome is closed and its size approaches a constant as more genomes are used, while for γ > 0, the pan-genome is open and its size increases as more genomes are included. As supported by the γ parameter from Heaps’ law, the *B. gladioli* pan-genome could be considered “open” state (γ = 0.37) with no sign of saturation, moderately expanding with the inclusion of new genomes [10]; whereas, the number of core genome and new genes remained relatively stable with the addition of new strains (Figure 3A,B). This phenomenon is common in species living in bacterial communities with large genomes and an open pan-genome [71]. For example, *B. pseudomallei* [72] and *B. cepacia* complex [73] genomes, which consist of large multiple circular chromosomal replicons, containing twice the amount of genetic material as *Escherichia coli*, are widely considered to have the “open” pan-genome. However, genomes of bacteria, which thrive only in a narrow range of niches, usually have a small size and “close” pan-genome [71,74].

The comparison of the protein-coding genes revealed that *B. gladioli* pan-genome contained a total repertoire of 11,403 gene clusters. The gene clusters in the accessory genome contributed a larger part of the pan-genome composition (6667 gene cluster, 58.5%) than those in the core genome (4736 gene clusters, 41.5%) (Figure 3C). These accessory genes were further classified into 3598 gene clusters in dispensable genome (shared by 2 to 13 strains), and 3069 gene clusters, which spread exclusively along independent strains as a unique genome. The accessory genome for gladiolus isolates (hereinafter “gladiolus-associated genome”) comprised 729 gene clusters that were missing from rice, human, and nature isolates. The accessory genomes for rice (hereinafter “rice-associated genome”), human (hereinafter “human-associated genome”), and nature (hereinafter “nature-associated genome”), which were not shared by isolates in other habitats, harbored 879, 735, and 1203 gene clusters. The Anvi’o program [47] was used to visualize the pan- and core genomes (Figure 3D). The Anvi’o visualization also highlighted the protein sets available in each one of the *B. gladioli* isolates and support genome conservation and differentiation. Below, we continuously investigated core genomes (universally conserved *B. gladioli* isolates) and the niche-associated genomes of gladiolus, rice, human, and nature isolates, which can contribute to adaptation in their respective habitat.

To assess the level of functional diversity within the *B. gladioli* core and accessory genomes, the COG analysis was employed (Table 3 and Appendix A). From 11,403 gene clusters, 6673 (59%) gene clusters were annotated in the COG functional categories; of these, 4736 gene clusters were part of the core genome shared by all 14 *B. gladioli*, and 2738 gene clusters were part of the accessory genome. The enrichment of COG category analysis presented that the gene clusters in the *B. gladioli* core genome were enriched for genes involved in COG class J (Translation, ribosomal structure and biogenesis), and class F (Nucleotide transport and metabolism) (Table 3). Genes composing accessory genomes were highly enriched in class L (replication, recombination, and repair), class Q (secondary metabolite biosynthesis, transport and catabolism), class W (extracellular structures), class U (intracellular trafficking, secretion, and vesicular transport), class V (defense mechanisms), class I (lipid transport and metabolism) and poorly characterized in some classes. Two independent pan-genome analyses of *Burkholderia* species, which occupy a large variety of hosts similar to *B. gladioli* species, identified COG functional categories for core and dispensable genomes [15,75]. Accordingly, the COG functional categories enriched for both core genomes of four *Burkholderia* genus (*B. cenocepacia*, *B. thailandensis*, and *B. pseudomallei* [15], and that of *B. pseudomallei* strains [75]) were also common with that of *B. gladioli* in our study, presenting ‘translation, ribosomal structure, and biogenesis’, and ‘nucleotide transport and metabolism’. However, the COG enrichment patterns for accessory genome resulting from these two studies were completely different from our result. Thus, the COG analysis result highlights that core genes can perform fundamental housekeeping functions regardless of their isolation sources, while accessory genes influence adaptation and survival in their own niche. Additionally, considering the open state of *B. gladioli* pan-genome, the result of COG enrichment analysis for *B. gladioli* accessory genome, especially COG class L (replication, recombination, and repair) and class Q (secondary metabolites biosynthesis, transport, and catabolism), also supports the perspective that larger genomes tend to accumulate functions like secondary metabolism [76] to allow organisms to reach a higher degree of ecological diversification.

### 3.4. Functional Analysis for Core and Niche-Associated Genome

To further clarify biological functions of core and niche-associated *B. gladioli* genome, we first performed GO enrichment analysis using core genome (Appendix A). The top-ranked 30 GO terms (a ranking by *p*-value) were significantly enriched based on three classes (cellular component, molecular function, and biological process). The GO terms were mainly involved in non-metabolic functions as shown in the COG result (Table 3). The most significantly enriched GO terms were ribosome (GO:0005840) and translation (GO:0006412) and they were consistent with COG class J (translation, ribosomal structure, and biogenesis). We also identified fundamental cellular processes and signaling functions in core genomes such as cell division (GO:0051301), cell cycle (GO:0007049), bacterial-type flagellum-dependent cell motility (GO:0071973), chemotaxis (GO:0006935), and signal transduction (GO:0007165). These results showed the relevance to COGs (Appendix A), which were not significantly enriched but comprised the majority of the pan-genome as core genes, including genes for cell motility (66%), cell cycle control, cell division, chromosome partitioning (59%), and signal transduction mechanisms (61%). Consequently, GO enrichment analysis reveals fundamental cellular processes and signaling functions of the core genome of *B. gladioli*.

Subsequently, we compared the enriched KEGG pathways in niche-associated genomes to explore the difference between the *B. gladioli* strains in terms of their biological capabilities and to highlight particular niche adaptations (Figure 4). Notably, there were clear differences in functions that were uniquely conserved in each niche-associated genome. The gladiolus-associated genome uniquely presented categories related to metabolism (i.e., cysteine and methionine metabolism, and selenocompound metabolism) that were absent in other groups. Selenium is an essential micronutrient used by organisms across all three domains of life [77,78,79]. Plants can absorb selenium from soil and assimilate various selenocompounds like selenomethionine and selenocysteine [80]. Selenium that is assimilated by plants can contribute to stress tolerance, inhibition of ROS damage, promotion of plant growth, homeostasis of essential nutrient elements, and photosynthesis [81,82,83,84,85]. Moreover, many enzymes within both plant and bacteria require selenium and selenoamino acids for activities [86]. Hence, the competition between plant hosts and pathogen for limited selenium resources is a natural consequence to increase their fitness [87]. Taken together, this KEGG result indicates that selenium availability will play a uniquely critical role in determining who wins the battle between host gladiolus and pathogenic *B. gladioli*.

The rice-associated genome was heavily enriched in genes related to aromatic compounds, styrene, and aminobenzoate degradation pathways (Figure 4). Among various aromatic compounds, lignin is an abundant terrestrial aromatic macromolecule [88]. Once lignin is depolymerized to monomers and/or low-molecular-weight aromatics, surrounding bacteria assimilate them for carbon and energy sources through their well-adapted metabolic pathways [89]. Various studies have also reported *Burkholderia* harboring gene clusters for degradation of lignin-derived aromatics [90,91]; to date, the ability of *B. gladioli* to degrade aromatic compounds remains unclear. However, this KEGG analysis result strongly suggests that rice pathogenic *B. gladioli* have enough potential to effectively use the lignin-derived aromatic compounds. Additionally, the following facts describing: (1) the seed-borne nature of *B. gladioli* in rice as confirmed in several previous reports [25,92,93] and (2) lignin as a major component of rice husk [94] support our expectation. Interestingly, we also identified the degradation of aromatic compounds pathway in the nature-associated genome (Figure 4), and this result can be explained by a capacity of *B. gladioli* Coa14, which was isolated from the Coari Lake (Amazonas, Brazil), for using crude oil as the only carbon source [28]. The crude oil is a complex mixture of both hydrocarbons like aromatic hydrocarbons, and non-hydrocarbons can easily enter and accumulate in high concentrations in aquatic and terrestrial environments [95,96,97]. Therefore, it is possible that the degradation of aromatic compounds is not well observed at the *B. gladioli* species level but is a unique function of *B. gladioli* strains living in rice plants and in nature.

Genes composing the human-associated genome were highly distributed in mismatch repair, RNA degradation, DNA replication, homologous recombination, and biofilm formation—Vibrio cholerae of KEGG pathways (Figure 4). The mismatch repair is a system for identifying and repairing mispaired base during DNA replication and recombination, as well as repairing some forms of DNA damages during infection [98,99]. Upon infection, bacterial cell components stimulate pathogen recognition receptors in hosts, thereby provoking inflammation [100]. Pathogens experience DNA and RNA damage by chemical mediators of inflammation and require multiple DNA repair pathways for optimal survival in the stressful niche condition [98,100,101]. Conversely, this stress-induced repair system in the pathogen allows genetic diversification and accelerates the adaptation [102,103]. In several human pathogenic bacteria such as *Helicobacter*, *Streptomyces*, and *Neisseria* [104,105,106], it was reported that recombination occurs frequently and can promote their adaptation [107]. Zhou et al. also reported that recombination in *Burkholderia cepacia* complex (BCC), which causes diseases in cystic fibrosis with *B. gladioli*, significantly influences adaptation and diversification [73]. Besides, the biofilm formation pathway in the human-associated genome indicates the difficulty of *B. gladioli* to withstand several host-defense measures [108]. Therefore, the available evidence suggests that DNA repair, recombination, and formation of biofilm are possibly necessary for survival in the cystic fibrosis-niche when human pathogenic *B. gladioli* experience infection-associated DNA damage. However, the conservation of these biological functions remains to be seen in human isolates across different clades since the human isolates included in this study are specific to clade 3 (Figure 2B).

The nature-associated genome retained not only the degradation of aromatic compounds but also a few pathways (i.e., quorum sensing, plant-pathogenic interaction, bacterial secretion system, and lipopolysaccharide biosynthesis) (Figure 4). As mentioned above, a close phylogenomic relationship between rice and nature isolates (Figure 2), and sharing of degradation of the aromatic compound pathway with rice- and nature-associated genomes can provide evidence of the phenomenon that most gene transfer occurs between closely related organisms [109,110]. For example, comparative genome analysis of 144 prokaryotic genomes shows the frequent transfer of genetic information between closely related taxa or between bacteria inhabiting the same environment [110]. Consistent with this report, it is expected that some genes for plant-pathogen interaction can be transferred to soil-derived isolate *B. gladioli* MSMB1756 from close plant pathogenic bacteria. Alternatively, secretion system and lipopolysaccharide biosynthesis in the nature-associated genome (Figure 4) can be explained by their high variation among *Burkholderia*. Specifically, the type 4 secretion system (T4SS), which transports not only proteins but also nucleic acids [111], is known as the most versatile family of secretion systems as per Seo et al. who report high variability of T4SS among *Burkholderia* strains [16]. Additionally, there is considerable diversity in lipopolysaccharide (LPS) that is present on the bacterial surface and is produced by most gram-negative bacteria [112]. We identified that genes of the nature-associated genome were condensed in the biosynthesis of the S-layer of the cell envelope. From decades of research, it is evident that S-layers, which are closely associated with the LPS of the outer membrane, can remarkably differ even between closely related species [113]. Considering the enormous diversity of S-layer, S-layers can reflect the evolutionary adaptations of bacteria to natural habitats and provide bacteria with advantages in the selection of specific environmental and ecological conditions [114,115].

### 3.5. In Silico Analysis of Nonribosomal Peptide Synthetases

Contrary to primary metabolites, secondary metabolites are not required for normal cell growth or development but can offer a competitive advantage for survival and adaptation in its ecological niche [116]. Of secondary metabolites, nonribosomal peptide (NRP), which is synthesized by nonribosomal peptide synthetase (NRPS) working in a ribosome-independent assembly line fashion [117], has attracted intense interest by its antimicrobial activity [118,119]. In this regard, *Burkholderia*-produced NRP and NPRS BGCs have been recently explored in some *Burkholderia* members, including *B. gladioli* [32,39], *B. ambifaria* [120], *B. pseudomallei* [121], and *B. thailandensis* [122]. Hence, to explore NPRS BGCs of *B. gladioli* in this study, genome mining was conducted using antiSMASH. Analysis of 14 *B. gladioli* strain revealed a remarkable array of BGCs that encode for rhizomide A/B/C, sulfazecin, icosalide A/B, bongkrekic acid, aluminide, etnangien, enacyloxin IIa, ralsolamycin, micacocidin, and xenotetrapeptide (Figure 5). Compared with the distribution of BGCs in the previously studied *B. gladioli* [32], we also consistently identified the distribution of BGCs for bongkrekic and icosalide in genomes of *B. gladioli* within specific clades. As in the findings of Jones et al. [32], the BGC for bongkrekic was specifically contained in the genome of strain MSMB1756 in clade 1A and the genome of strain BSR3 in clade 1B. The BGC for icosalide was widely distributed in all 14 *B. gladioli* strains within clades 1A, 1B, 2, and 3 (Figure 2B and Figure 5). This result implies that some BGCs tend to be distributed in a clade-specific manner. We additionally detected several BGCs which were not investigated in other studies. Among them, rhizomide BGC was present in all the *B. gladioli* strains. The gladiolus and human isolates occupied additional rhizomide BGCs and particularly, human isolates, including AU0032, AU29541, and AU30473, were rich in rhizomide BGCs more than any other strains. By spectroscopic characterization, production of rhizomide was identified from soil bacterium *Burkholderiales* DSM 7029, and *Paraburkholderia rhizoxinica*, intracellular symbionts of the phytopathogenic fungus [123,124,125]. Additionally, *Burkholderia*-originated rhizomide exhibited various actives such as antitumor activity, protective activity against cucumber downy mildew, antibacterial activity for gram-positive bacteria [123]. Considering this, the conservation of rhizomide BGC in this study indicates that *B. gladioli*, especially that from human isolates, can also be a high potential producer for rhizomide. Interestingly, gladiolus isolates also uniquely conserved BGCs for etnangien, which is also well known for antibiotic activity against gram-positive bacteria by inhibiting nucleic acid polymerases [126]. An exception, ATCC 25417, BSR3, and KACC 18962 strains uniquely carried BGC for ralsolamycin, xenotetrapeptide, and enacycloxin IIa, respectively. It can be inferred from the exceptional acquisition of biosynthetic genes by the exchange of genetic information between contemporary organisms.

### 3.6. Prediction of CRISPR/Cas System

In the past, the most representative roles of CRISPR systems are protecting bacteria from invasion by bacteriophages or foreign plasmid DNA [127,128]. CRISPR systems have recently received attention for association with virulence and biofilm formation in pathogenic bacteria, and modulation of their expression by various environmental stimuli [129]. For example, it was reported that QS-dependent regulation of CRISPR/Cas genes influences the bacterial immune system in *B. plantarii* PG1 (formerly *B. glumae* PG1) [130] and *Pseudomonas aeruginosa* [131]. With this respect, analysis of the CRISPR/Cas system was performed for *B. gladioli* genomes (Table 4) to find a link between the CRISPR/Cas system and the different surrounding environments of *B. gladioli*. CRISPRs were identified in most *B. gladioli* genomes except in AU29541, FDAARGOS_390, and FDAARGOS_391 strains, and the numbers of CRISPR arrays in one genome varied from one to four. It was assumed that these CRISPRs could not silence foreign DNA, because there were many orphan CRISPR arrays without associated Cas proteins. One interesting fact about these orphan CRISPR arrays is that spacer sequences of BSR3 matched to plasmid sequences of *Cupriavidus taiwanensis*. *C. taiwanensis* (originally called *Ralstonia taiwanensis*) is a nitrogen-fixing bacterium of the family Burkholderiaceae, forming root nodules on host plant [132]; meanwhile, *C. taiwanensis* were also isolated from the sputum of a cystic fibrosis patient and root nodule [133]. Additionally, the coexistence of *Burkholderia* and *Cupriavidus* has frequently been observed in plant roots and rhizosphere [134,135,136]. Therefore, the presence of spacer in BSR3 matching to *C. taiwanensis* indicates traces of interaction between *Burkholderia* and *Cupriavidus* not only in plant roots but also in the sputum of a cystic fibrosis patient and provisionally indicates cross-kingdom infection between plant and human [137,138]. Whereas, two *B. gladioli* strains (Coa14 and MSMB1756 of nature isolates) contained complete CRISPR/Cas systems and all those systems affiliated with type I-F that contained Cas proteins including Cas1, Cas3-Cas2, Csy1 to Csy3, and Cas6. In *P. aeruginosa* PA14, the type I-F CRISPR/Cas system functioned to eliminate invading DNA not RNA [139,140] and Buyukyoruk et al. also demonstrated an efficient barrier role of the type I-F CRISPR system by restricting dsDNA [141]. Ironically, we confirmed the spacer sequences of both Coa14 and MSMB1756 strains commonly matched to plasmid sequences of *Streptomyces* species (*S. lavendulae*, *S. aureofaciens*, *S. rimosus*, and *S. katrae*) and *Ralstonia solanacearum* (Table 4). *Streptomyces* members are widely considered the source of many tetracycline antibiotics and an important reservoir of antibiotic resistance genes in soil [142,143,144,145]. *R. solanacearum* is the most destructive phytopathogen that attacks many crops and other plants over a broad geographical range [146,147]. It is obvious that such antibiotic resistance and virulence genes in bacterial plasmids are important fitness traits that increase the survival ability of bacteria within the host and in nature. However, we couldn’t identify any antibiotic resistance genes or virulence-related genes in spacer sequences of both Coa14 and MSMB1756 strains. We speculate that the existence of sequences of *Streptomyces* and *R. solanacearum* in the spacer of *B. gladioli* was just vestiges of the interaction between distinct bacterial genera in nature. Furthermore, we suppose that the CRISPR/Cas system of *B. gladioli* may be an obstacle to acquiring antibiotic resistance and virulence genes. This inverse relationship between CRISPR/Cas and antibiotic resistance has been reported in *Enterococcus* species [148]. It was confirmed that a strong correlation between the absence of complete CRISPR-Cas loci and the emergence of multidrug-resistant enterococcal strains by analyzing a large genome data set. Hence, it is reasonable that none of the plant and human pathogenic *B. gladioli* have a complete and active CRISPR/Cas system.

## 4. Conclusions

Our pan-genomic study investigated the relationship between fourteen *B. gladioli* genomes and the ecological niche of their isolation. Specifically, we tried to prove the association between the bacteria’s phylogenomic distribution and habitat, and to find possible biological functions that could be shared by each niche-associated isolate. The pan-genomic analysis revealed that the enriched function of selenium availability in gladiolus pathogenic *B. gladioli* tends to be a uniquely important role during the competition between plant host and pathogen for the limited selenium resources. The aromatic compound degradation pathway in the genomes of *B. gladioli* isolates, which originate from rice and nature, was indicative of seed-borne nature and crude oil-accumulated nature, respectively. Enriched functions of human-associated genomes implied the possibility of promoting genetic diversification and accelerating the adaptation by stress-induced DNA repair systems and recombination, and of the possible protective role of biofilm formation as the host-defense measure in the cystic fibrosis-niche. We also identified the conservation of rhizome BGC in all *B. gladioli* strains and concentrated distribution of this cluster in human isolates. Finally, we confirmed the absence of a complete CRISPR/Cas system in both plant and human pathogenic *B. gladioli* and the presence of the system in *B. gladioli* living in nature, possibly reflecting the inverse relationship between CRISPR/Cas system and antibiotic resistance. Thus, this framework will provide snapshots of how ecological niches affect the versatility of gene contents in the genome and contribute to the metabolic diversity of bacteria.

## Figures and Tables

**Figure 1 microorganisms-09-01123-f001:**
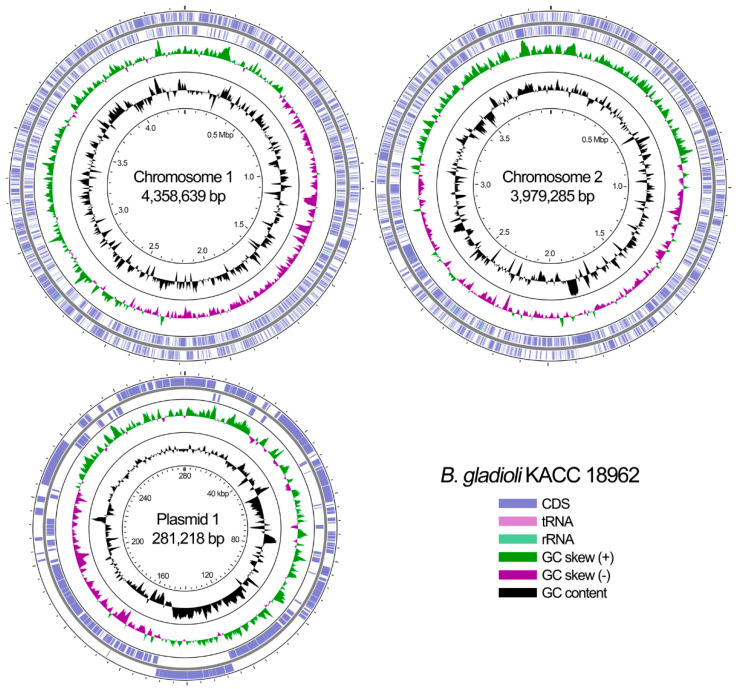
Circular view of complete genome of rice pathogenic *Burkholderia gladioli* KACC 18962. The key describes the single circles in the top-down outermost-innermost direction. The circular maps were generated by CGView [68].

**Figure 2 microorganisms-09-01123-f002:**
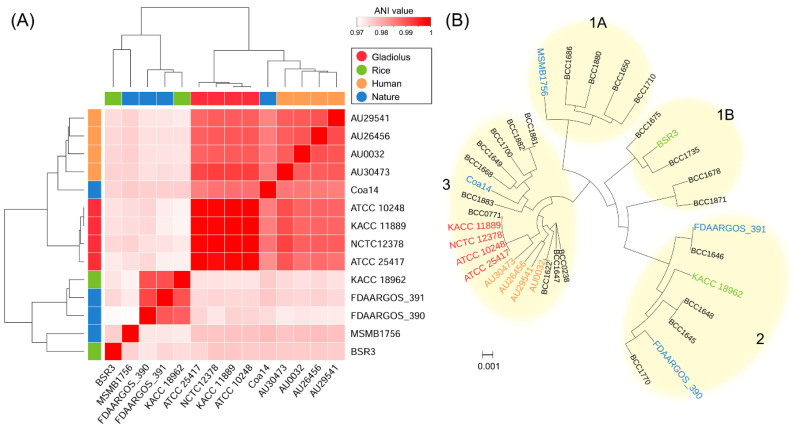
Phylogenomic distribution of *Burkholderia gladioli* isolates. (**A**) Comparison of the average nucleotide identity (ANI) values between each genome of the 14 strains of *B. gladioli*. Heatmaps display the percentage average nucleotide identities between the 14 *B. gladioli* strains. Color bars above and to the left of the heatmaps correspond to isolation source of each *B. gladioli* strain. The color keys represent the identity of strains with lower (white), and higher (red) ANI values. Hierarchal cluster dendrograms were generated based on similar ANIb values across each strain. The ANI was calculated by the Python3 module pyani using blastn alignment. (**B**) Core genome phylogeny of *B. gladioli* isolates. A total of 4392 core genes were identified using 14 *B. gladioli* strains which are listed in (**A**) and 22 *B. gladioli* isolates, which were analyzed in previous studies [39]. The concatenated core genes were aligned and used to construct phylogeny with 1000 rapid bootstraps. Classification of clades 1A, 1B, 2, and 3 was adopted from the previous study by Jones et al. [32]. The scale bar displays the number of base substitutions per site.

**Figure 3 microorganisms-09-01123-f003:**
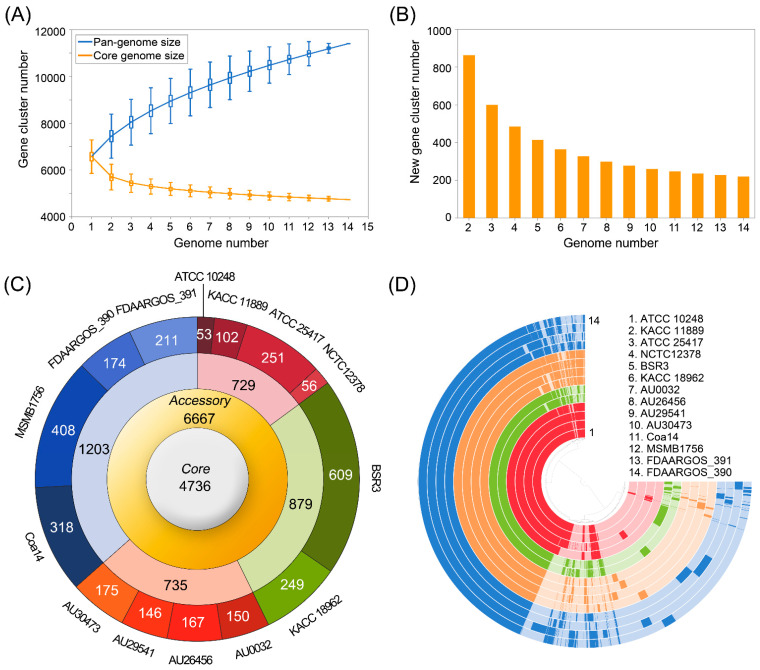
Pan-genome analysis of *Burkholderia gladioli* strains (**A**) Pan- and core genome size accumulation. The colored boxes respectively denote the pan-genome (blue) and core genome (orange) sizes for each genome for comparison. The curve is the least square fit of the power law for the average values. (**B**) New gene family accumulation. The orange bars illustrate the number of expected new genes detected with every increase in the number of *B. gladioli* genomes. (**C**) Distribution of gene cluster numbers in the *B. gladioli* pan-genome. A circular plot schematic representation depicts the number of gene clusters in core, accessory, dispensable, and unique genomes. From inside to outside: Core genome, accessory genome, niche-associated accessory genome, which is further divided into gladiolus-associated (red), rice- associated (green), human- associated (orange), and nature- associated (blue) genome, and unique genome at the strain level. (**D**) Comparative overview of pan- and core genomes obtained from the Anvi’o tool. The different *B. gladioli* isolate groups are displayed by red (gladiolus isolates), green (rice isolates), orange (human isolates), and blue (nature isolates). Each track indicates a genome which is presented by number and color variations (dark/light) shows presence/absence of genes per genome.

**Figure 4 microorganisms-09-01123-f004:**
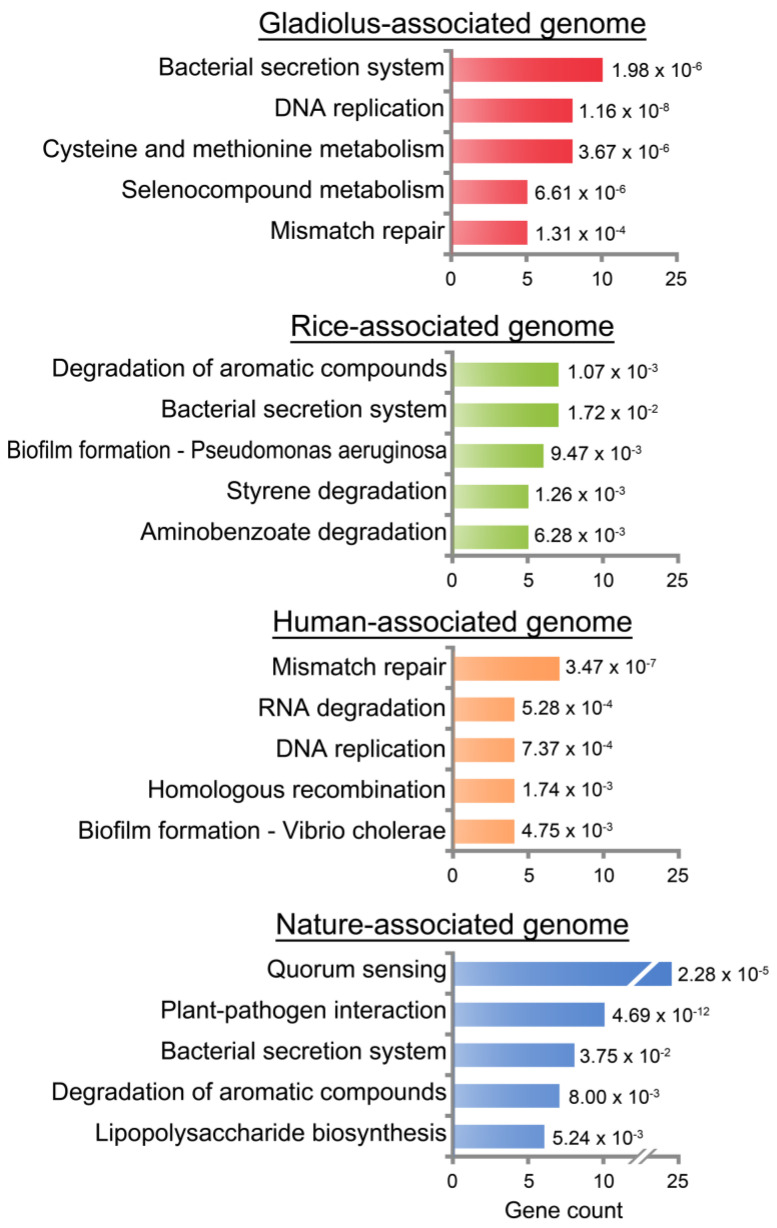
KEGG enrichment analysis of *Burkholderia gladioli* niche-associated genomes. The gladiolus-associated genome was derived from gladiolus isolates (ATCC 10248, KACC 11889, ATCC 25417, and NCTC 12378), the rice- and human-associated genomes were derived from rice isolates (BSR3 and KACC 18962), and human isolates (AU0032, AU26456, AU29541, and AU30473), respectively. The nature-associated genome was derived from nature isolates (Coa14, MSMB1756, FDAARGOS_390, and FDAARGOS_391). The *y*-axis represents enriched KEGG pathways. The *x*-axis shows the count of genes in each niche-associated genome. KEGG pathways with a *p*-value < 0.05 were considered significantly enriched. *P*-value from the hypergeometric test is displayed next to each bar.

**Figure 5 microorganisms-09-01123-f005:**
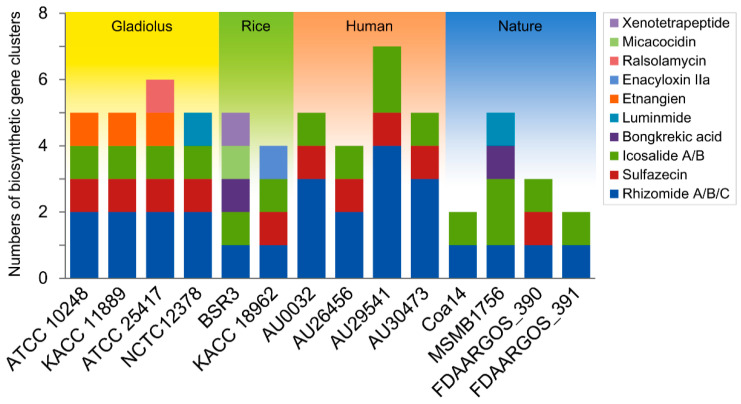
The number of nonribosomal peptide synthetase (NRPS) genes predicted from *Burkholderia gladioli* genomes. NPRS biosynthetic gene cluster (BGC) in genomes of 14 *B. gladioli* strains were identified thorough antiSMASH.

**Table 1 microorganisms-09-01123-t001:** Genome statistics of *Burkholderia gladioli* KACC 18962.

Feature	Chromosome 1	Chromosome 2	Plasmid 1
Genome sequencing level	Complete		
Total number of reads	10,350,496		
Total yield (bp)	1,562,924,896		
Bases with a phred value > 20 (%)	92.02		
Bases with a phred value > 30 (%)	82.95		
Sequencing depth	190	187	190
Genome size (bp)	4,358,639	3,979,285	281,218
Genome G + C content (%)	67.56	68.62	61.1
No. of genes	3982	3250	243
No. of coding genes	3820	3169	227
No. of pseudogenes	93	65	15
No. of RNA genes (16S/5S/23S)	3/3/3	2/2/2	0
No. of tRNA genes	56	10	1
Other RNA	4	0	0

**Table 2 microorganisms-09-01123-t002:** Genomes information of the 14 strains of *Burkholderia gladioli* used in this study.

*B. gladioli* Strains	Source	Assembly	Size (Mb)	GC (%)	Level	Proteins
ATCC 10248	Gladiolus	GCA_000959725.1	8.9	67.6	Complete	7514
KACC 11889	Gladiolus	GCA_002208175.1	8.9	67.7	Complete	7221
ATCC 25417	Gladiolus	GCA_000756855.1	9.3	67.3	Scaffold	7941
NCTC 12378	Gladiolus	GCA_900446225.1	8.4	68.0	Contig	7098
BSR3	Rice	GCA_000194745.1	9.1	67.4	Complete	7639
KACC 18962	Rice	This study	8.6	67.8	Complete	7216
AU0032	Sputum ^a^	GCA_002980975.1	8.0	68.2	Contig	6769
AU26456	Sputum	GCA_002981405.1	8.1	68.2	Contig	6874
AU29541	Sputum	GCA_002981475.1	8.4	68.2	Contig	7112
AU30473	Sputum	GCA_002981875.1	8.1	68.3	Contig	6928
Coa14	Water	GCA_002917905.1	8.5	68.0	Contig	7189
MSMB1756	Soil	GCA_001527485.1	8.2	68.1	Contig	6946
FDAARGOS_390	Nature ^b^	GCA_002554225.1	8.8	67.6	Contig	7470
FDAARGOS_391	Nature	GCA_002554395.1	8.4	68.0	Contig	7058

^a^*Homo sapiens*; ^b^ unspecified in NCBI database.

**Table 3 microorganisms-09-01123-t003:** Clusters of orthologous groups (COG) enriched in core and dispensable genomes of *Burkholderia gladioli*.

Core	COG	No. Cluster	*p*-Value
Class			
J	Translation, ribosomal structure and biogenesis	172	1.52 × 10^−4^
F	Nucleotide transport and metabolism	91	4.79 × 10^−4^
Dispensable			
Class	COG	No. cluster	*P*-value
L	Replication, recombination and repair	234	0
Q	Secondary metabolites biosynthesis, transport and catabolism	297	0
-	Unclassified	3929	0
W	Extracellular structures	29	5.52 × 10^−12^
U	Intracellular trafficking, secretion, and vesicular transport	147	6.91 × 10^−7^
V	Defense mechanisms	53	6.36 × 10^−4^
R	General function prediction only	541	3.76 × 10^−3^
I	Lipid transport and metabolism	176	1.27 × 10^−2^

**Table 4 microorganisms-09-01123-t004:** Predicted CRISPR/Cas system in *Burkholderia gladioli*.

Strain	CRISPR Count	Type	Cas Gene Count	Cas Gene	No. Spacers	Spacer Match
ATCC 10248	4	-	-	-	1,1,1,1	ND ^a^
KACC 11889	4	-	-	-	1,1,1,1	ND
ATCC 25417	2	-	-	-	1,1	ND
NCTC 12378	4	-	-	-	1,1,1,1	ND
BSR3	2	-	-	-	1,2	*Cupriavidus taiwanensis*
KACC 18962	1	-	-	-	1	ND
AU0032	1	-	-	-	1	ND
AU26456	2	-	-	-	1,1	ND
AU29541	-	-	-	-	-	-
AU30473	2	-	-	-	1,1	ND
Coa14	3	Type I-F	6	Cas1,Cas3-Cas2,Csy1,Csy2,Csy3,Cas6	1,1,39	*Streptomyces* spp. ^b^ *Ralstonia solanacearum*
MSMB1756	2	Type I-F	6	Cas1,Cas3-Cas2,Csy1,Csy2,Csy3,Cas6	1,46	*Streptomyces* spp. *Ralstonia solanacearum*
FDAARGOS_390	-	-	-	-	-	-
FDAARGOS_391	-	-	-	-	-	-

^a^ Not determined; ^b^
*S. lavendulae*, *S. aureofaciens*, *S. rimosus*, and *S. katrae*.

## Data Availability

The genome sequences of *B. gladioli* KACC 18962 chromosome 1, chromosome 2, and plasmid genome are available in the NCBI GenBank under accession numbers CP045573, CP045574, and CP045575, respectively.

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
