# Peer review of "Pan-Genome Analysis Reveals Host-Specific Functional Divergences in Burkholderia gladioli"

_microorganisms, 2021, doi:10.3390/microorganisms9061123_

Round 1
Reviewer 1 Report
The manuscript submitted by Lee et. al. is interesting and very well presented. They aim to define niche specific genomes, and from that inferred niche specific functions, from the multifaceted bacterium Burkholderia gladioli. They present the sequence of a novel isolate from Rice and perform a broad ranging comparative analysis of a further 13 sequenced B. gladioli isolated from diverse niches (humans, gladiolus, rice, and ‘nature’). Bioinformatic analyses are employed to identify niche specific genes and gene clusters, and a systematic analysis of niche specific traits is presented. They attempt to explore the highly relevant question of how one pathogen is capable of success in such a range of diverse niches, and to uncover the genetic basis of such adaptation.
While the analyses presented appear technically sound and very well written, I must raise the following concerns about on the validity of the conclusions of the paper and their broad application to B. gladioli as a species:
- A major limitation of the study is the inclusion of just 14 representative genomes of B. gladioli. The authors state in their introduction that over 200 genomes for this species are available. Do the authors agree that expanding their dataset would give a better representation of the species and solidify their findings? Clearly the tools are available to perform a much more comprehensive analysis. It would be important to see if the ‘niche-specific’ conclusions hold true when a more comprehensive collection of isolates is included in the analysis.
- Such a small number of genomes may not cover the full diversity of this species, which may affect the outcomes of the analyses. A more complete analysis of population diversity is available (Jones et. al. 2021). Do the authors understand how well their 14 isolates capture the known diversity of the species, and do they have representatives from each phylogenomic group? I bring the authors attention to Lin et. al. (2021) who elegantly map their isolate within the population diversity of the species using a subset of isolates (Figure 3). I think the current manuscript would benefit from similar analyses to indicate how well their chosen isolates cover the full diversity of the population, which would help with the application of their findings to the full species.
- The niche-based understanding of B. gladioli population biology may not be wholly appropriate and may be leading to false findings on their small collection of isolates. For example, the rice niche has only two representative isolates. I do not believe that any core or accessory genome can confidently be concluded from so few isolates. Is it possible to confirm the findings by the inclusion of more representative isolates from each niche? A better understanding of the population diversity covered (point 2) and a larger subset of isolates (point 1) will help address this point.
- Further niche-based conclusions are made for suggested human adapted B. gladioli (n=4). Work by Jones et. al. (2021) indicates there is no human adapted lineage of B. gladioli, and in fact, isolates from Cystic Fibrosis are found in all phylogenomic clades identified. There was no phylogenetic split between plant pathogenic B. gladioli and human pathogenic B. gladioli. The suggestion is that B. gladioli CF infection originate from independent acquisition of environmental strains. This raises questions about the validity of a human clade specific genome of B. gladioli identified by the authors and needs to be addressed. My feelings are that the small number of human isolates included is a vast underestimation of the diversity of B. gladioli capable of CF infection, and the conclusions to a ‘human niche’ are inappropriate. Can the authors explain how their niche designations and previous phylogenomic clusters relate to each other, and how their niche designations and historic pathogen designations for B. gladioli relate to each other?
- Four isolates belong to a niche described as ‘Nature’. This grouping seems vague. What do the authors describe as ‘Nature’ and is this a valid, single, niche? Can the isolation source of these isolates be described any more specifically?
- Niche specific genes, clusters and functional conclusions are therefore overstated in this paper, given the problems with niche definition and full coverage of the diversity. There are conclusions that the presence of specific enriched genes indicates a precise function. These conclusions should be down stated. The presence of genes/clusters indicates a possible function, but each function would need to be experimentally validated, and more caution shown in this manuscript with inferring function from genome data. I am not suggesting experimental validation must be performed for this manuscript, only that the need for this should be clearly stated when making conclusions.
- For section 3.5, the published findings of Jones et al. (2021) on the distribution of secondary metabolite clusters in B. gladioli should be compared and discussed.
- Minor points: The Comparison of the average nucleotide identity figure is incorrectly numbered. The number of isolates included in each specific genome analysis should be indicated in figure 4. A rationale for exploring Crispr-Cas systems should be included.
Line 546 – given the previous points raised in this review, I cannot agree that the establishment of relationships between the bacteria’s phylogenomic distributions and habitats has been proved.
Jones, C., Webster, G., Mullins, A., Jenner, M., Bull, M., Dashti, Y., Spilker, T., Parkhill, J., Connor, T., LiPuma, J., Challis, G. and Mahenthiralingam, E., 2021. Kill and cure: genomic phylogeny and bioactivity of Burkholderia gladioli bacteria capable of pathogenic and beneficial lifestyles. Microbial Genomics, 7(1).
Lin, Y., Lee, C., Leu, W., Wu, J., Huang, Y. and Meng, M., 2021. Fungicidal Activity of Volatile Organic Compounds Emitted by Burkholderia gladioli Strain BBB-01. Molecules, 26(3), p.745.
Author Response
Response to Reviewer 1 Comments
Point 1: A major limitation of the study is the inclusion of just 14 representative genomes of B. gladioli. The authors state in their introduction that over 200 genomes for this species are available. Do the authors agree that expanding their dataset would give a better representation of the species and solidify their findings? Clearly the tools are available to perform a much more comprehensive analysis. It would be important to see if the ‘niche-specific’ conclusions hold true when a more comprehensive collection of isolates is included in the analysis.
Response 1: We agree with the reviewer that inclusion of hundreds of genomes in a pan-genomic study could give better identification of the diversity and composition of the global gene repertoire. As the reviewer mentioned in point 4, we also previously confirmed that dispersed distribution of isolates from cystic fibrosis in phylogenomic tree. Nevertheless, we aimed in this study that to find as many biological functions as possible that are common in bacteria sharing same habitat, because there are several studies indicating the impact of environmental variability in the evolution of microbial genomes such as phylogenetic position and the diversity of metabolic capabilities [1,2]. It is also supposed that bacteria with high genome sequence similarity are more likely to discover common biological functions providing adaptability, pathogenicity, and stress responses for survival in particular habitat.
In this respect, it was difficult to analysis the pan-genome that was constructed using entire hundred genomic sequences of B. gladioli in identifying meaningful functions; thus, we selected a total of 14 B. gladioli strains, to the best of our knowledge, which have been historically, phenotypically, or genomically well characterized of Burkholderia members. We have discussed the reason for selection of 14 B. gladioli strains in the Results and Discussion section (page 5, lines 208–221).
Furthermore, according to a study finding the minimum number of genomes necessary for a comprehensive analysis, pan-genomic analysis of each 27 bacterial species (e.g., Escherichia coli, Bacillus cereus, and Pseudomonas aeruginosa etc.), ten genomes can be sufficient in large and opened pan-genome for reliable analysis [3]. Vernikos et al. also suggested that a pan-genome analysis even with extreme sampling can be reliable bypassing the need to follow an exhaustive all-against-all comparison [4,5]. On that basis, we think that construction of the pan-genome in our study can be reliable to provide snapshot view of the functional diversity of B. gladioli in accordance with habitat.
Related reference:
- Dutta, C.; Paul, S. Microbial lifestyle and genome signatures. Curr. Genomics 2012, 13, 153–162, doi:10.2174/138920212799860698.
- diCenzo, G.C.; Finan, T.M. The divided bacterial genome: Structure, function, and evolution. Microbiol. Mol. Biol. Rev. 2017, 81, doi:10.1128/mmbr.00019-17.
- Rouli, L.; Merhej, V.; Fournier, P. E.; Raoult, D. The bacterial pangenome as a new tool for analysing pathogenic bacteria. New Microbes New Infect. 2015, 7, 72–85, doi:10.1016/j.nmni.2015.06.005.
- Vernikos, G.S. A review of pangenome tools and recent studies, in The Pangenome: Diversity, Dynamics and Evolution of Genomes, eds Tettelin, H and Medini, D (Cham: Springer), 2020, 89-112, doi: 10.1007/978-3-030-38281-0_4.
- Vernikos, G.S.; Medini, D; Riley, D.R.; Tettelin, H.; Ten years of pan-genome analyses. Curr. Opin. Microbiol. 2015, 23, 148–154, doi:10.1016/j.mib.2014.11.016.
Point 2: Such a small number of genomes may not cover the full diversity of this species, which may affect the outcomes of the analyses. A more complete analysis of population diversity is available (Jones et. al. 2021). Do the authors understand how well their 14 isolates capture the known diversity of the species, and do they have representatives from each phylogenomic group? I bring the authors attention to Lin et. al. (2021) who elegantly map their isolate within the population diversity of the species using a subset of isolates (Figure 3). I think the current manuscript would benefit from similar analyses to indicate how well their chosen isolates cover the full diversity of the population, which would help with the application of their findings to the full species.
Response 2: As we responded point 1, we have addressed and discussed the reason for selecting 14 B. gladioli strains using the pan-genomic analysis and their representatives in the Results and Discussion section (page 5, lines 208–221).
We would like to thank the reviewer for introducing us to such analysis for covering the diversity of the population. We carefully prospected the studies by Jones et al. [1] and Lin et al. [2], and applied their analyses to our study with the reference. The detailed method was addressed in the Materials and Methods (page 3, lines 125–135) and additionally changed the section title to ‘Phylogenomic analysis’ (page 3, line 120). The result was presented as Figure 2B and the relevant legend was added (page 9, lines 302–311). We have also discussed in the section 3.2. Phylogenomic analysis (page 8, lines 288–300). Shortly, we obtained 4392 concatenated core genes by comparing 14 B. gladioli genomes in this study with the genomes of 22 previously reported by Lin et al [1]. As a result, both gladiolus and human isolates were positioned in clade 3. The rice isolate BSR3 and nature isolate MSMB1756 were solely belonged to clade 1B and 1A, respectively. The rice isolate KACC 18962 was classified into clade 2 with FDAARGOS_390, and FDAARGOS_391 of nature isolates. We confirmed the rest 22 B. gladioli isolates were positioned identically as the previous classification [1]. Consequently, this cladistic structure was consistent and similar with the ANI-based phylogenomic relationships in Figure 2A.
Related reference:
- Jones, C.; Webster, G.; Mullins, A.J.; Jenner, M.; Bull, M.J.; Dashti, Y.; Spilker, T.; Parkhill, J.; Connor, T.R.; Lipuma, J.J.; et al. Kill and cure: Genomic phylogeny and bioactivity of Burkholderia gladioli bacteria capable of pathogenic and beneficial lifestyles. Microb. Genomics 2021, 7, 1–13, doi:10.1099/mgen.0.000515.
- Lin, Y.T.; Lee, C.C.; Leu, W.M.; Wu, J.J.; Huang, Y.C.; Meng, M. Fungicidal activity of volatile organic compounds emitted by Burkholderia gladioli Strain BBB-01. Molecules 2021, 26, 745, doi.org/10.3390/molecules26030745.
Point 3: The niche-based understanding of B. gladioli population biology may not be wholly appropriate and may be leading to false findings on their small collection of isolates. For example, the rice niche has only two representative isolates. I do not believe that any core or accessory genome can confidently be concluded from so few isolates. Is it possible to confirm the findings by the inclusion of more representative isolates from each niche? A better understanding of the population diversity covered (point 2) and a larger subset of isolates (point 1) will help address this point.
Response 3: We firstly thank the reviewer for suggestion about including more representative isolates from each niche. However, as we responded point 1, it was difficult to include more B. gladioli genomes in our pan-genomic analysis because our intention was to show the common biological functions presented by each niche-associated B. gladioli genome. Instead of this, we performed phlyogenomic analysis as the reviewer suggested in point 2. We thank the reviewer again for giving us the opportunity to improve our study.
In addition, it is worth noting that the genomes of two rice isolates, B. gladioli BSR3 and B. gladioli KACC 18962 are only completely sequenced rice pathogenic B. gladioli genomes with good quality. The use of complete genome sequences with good quality can be one of the key factors allowing reliable pan-genomic analysis [1–3]. Especially, B. gladioli BSR3 is also one of the most well known and characterized rice pathogenic B. gladioli [4–7]. According to this, we needed to analysis rice isolates in this study.
Related reference:
- Rouli, L.; Merhej, V.; Fournier, P. E.; Raoult, D. The bacterial pangenome as a new tool for analysing pathogenic bacteria. New Microbes New Infect. 2015, 7, 72–85, doi:10.1016/j.nmni.2015.06.005.
- Vernikos, G.S. A review of pangenome tools and recent studies, in The Pangenome: Diversity, Dynamics and Evolution of Genomes, eds Tettelin, H and Medini, D (Cham: Springer), 2020, 89-112, doi: 10.1007/978-3-030-38281-0_4.
- Vernikos, G.S.; Medini, D; Riley, D.R.; Tettelin, H.; Ten years of pan-genome analyses. Curr. Opin. Microbiol. 2015, 23, 148–154, doi:10.1016/j.mib.2014.11.016.
- Seo, Y.S.; Lim, J.; Choi, B.S.; Kim, H.; Goo, E.; Lee, B.; Lim, J.S.; Choi, I.Y.; Moon, J.S.; Kim, J.; et al. Complete genome sequence of Burkholderia gladioli BSR3. J. Bacteriol. 2011, 193, 3149.
- Fory, P.A.; Triplett, L.; Ballen, C.; Abello, J.F.; Duitama, J.; Aricapa, M.G.; Prado, G.A.; Correa, F.; Hamilton, J.; Leach, J.E.; et al. Comparative analysis of two emerging rice seed bacterial pathogens. Phytopathology 2014, 104, 436–444, doi:10.1094/PHYTO-07-13-0186-R.
- Naughton, L.M.; An, S. qi; Hwang, I.; Chou, S.H.; He, Y.Q.; Tang, J.L.; Ryan, R.P.; Dow, J.M. Functional and genomic insights into the pathogenesis of Burkholderia species to rice. Environ. Microbiol. 2016, 18, 780–790, doi:10.1111/1462-2920.13189.
- Kim, S.; Park, J.; Choi, O.; Kim, J.; Seo, Y.S. Investigation of quorum sensing-dependent gene expression in Burkholderia gladioli BSR3 through RNA-seq analyses. J. Microbiol. Biotechnol. 2014, 24, 1609–1621, doi:10.4014/jmb.1408.08064.
Point 4: Further niche-based conclusions are made for suggested human adapted B. gladioli (n=4). Work by Jones et. al. (2021) indicates there is no human adapted lineage of B. gladioli, and in fact, isolates from Cystic Fibrosis are found in all phylogenomic clades identified. There was no phylogenetic split between plant pathogenic B. gladioli and human pathogenic B. gladioli. The suggestion is that B. gladioli CF infection originate from independent acquisition of environmental strains. This raises questions about the validity of a human clade specific genome of B. gladioli identified by the authors and needs to be addressed. My feelings are that the small number of human isolates included is a vast underestimation of the diversity of B. gladioli capable of CF infection, and the conclusions to a ‘human niche’ are inappropriate. Can the authors explain how their niche designations and previous phylogenomic clusters relate to each other, and how their niche designations and historic pathogen designations for B. gladioli relate to each other?
Response 4: Four human isolates (AU0032, AU26456, AU29541 and AU30473), opportunistic human pathogen occupying multidrug-resistance, have contained in the Burkholderia cepacia Research Laboratory and Repository strain collection [1,2]. Theses isolates were firstly reported and studied by Zeiser et al. [2,3] in order to susceptibility testing with various antibiotics.
We agree again the reviewer's and other research teams’ point of view that dispersed distribution of isolates from cystic fibrosis in all phylogenomic clades. We also recognized that there were overstatements related to split between gladioli and human isolates when discussed the result about phylogenomic analysis; thus, we have rewritten section 3.2. Phylogenomic analysis (page 8, lines 270–282). Additionally, it was identified that four human isolates continuously showed close relationships with each other both in whole genome-based and core genome-based phylogenetic trees, although gladiolus and human isolates were also closely positioned in clade 3 (Figure 2A and B).
Related reference:
- Papp‐Wallace K.M.; Becka S.A.; Zeiser E.T.; Ohuchi, N.; Mojica, M.F.; Gatta, J.A.; et al. Overcoming an extremely drug resistant (XDR) pathogen: Avibactam restores susceptibility to ceftazidime for Burkholderia cepacia complex isolates from cystic fibrosis patients. ACS Infect. Dis. 2017, 3, 502‐ 511, doi.org/10.1021/acsinfecdis.7b00020.
- Zeiser E.T.; Becka S.A.; Barnes M.D.; Taracila M; LiPuma J.J.; Papp-Wallace K.M. “Resurrecting old β-lactams”: the potent inhibitory activity of temocillin against multidrug resistant Burkholderia spp. isolates from the United States. Antimicrobial agents and chemother. 2019, 63, 4, doi.org/10.1128/AAC.02315-18.
- Zeiser E.T.; Becka S.A.; Wilson B.M.; Barnes M.D.; LiPuma J.J.; Papp‐Wallace K.M. "Switching partners": piperacillin‐avibactam is a highly potent combination against multidrug‐resistant Burkholderia cepacia complex and Burkholderia gladioli cystic fibrosis isolates. J. Clin. Microbiol. 2019, doi.org/10.1128/JCM.00181-19.
Point 5: Four isolates belong to a niche described as ‘Nature’. This grouping seems vague. What do the authors describe as ‘Nature’ and is this a valid, single, niche? Can the isolation source of these isolates be described any more specifically?
Response 5: As suggested by the reviewer, we more specifically stated sources of the B. gladioli (Coa14, and MSMB1756) in Table 2. Unfortunately, we tried hard to find specific source of strains FDAARGOS_390 and FDAARGOS_390 in the NCBI database and in literatures, but we could not find it. Thus, we have specified following text in the footer of Table 2: ‘b unspecified in NCBI database’.
Point 6: Niche specific genes, clusters and functional conclusions are therefore overstated in this paper, given the problems with niche definition and full coverage of the diversity. There are conclusions that the presence of specific enriched genes indicates a precise function. These conclusions should be down stated. The presence of genes/clusters indicates a possible function, but each function would need to be experimentally validated, and more caution shown in this manuscript with inferring function from genome data. I am not suggesting experimental validation must be performed for this manuscript, only that the need for this should be clearly stated when making conclusions.
Response 6: We changed the overstatement throughout the manuscript and in Figure 4 as follows: ‘niche-specific’ to ‘niche-associated’, ‘gladiolus-specific’ to ‘gladiolus-associated’, ‘rice-specific’ to ‘rice-associated’, ‘human-specific’ to ‘human-associated’, and ‘nature-specific’ to ‘nature-associated’. We also downstated in section of the Abstract (page 1, lines 14–15, line 19, line 21), the Results and discussion (page 5, lines 204–207), 3.4 Functional analysis for core and niche-associated genome (page 14, lines 463–465, lines 482-485, lines 490-492), and the Conclusion (page 17, line 603–page 18, line 620).
Point 7: For section 3.5, the published findings of Jones et al. (2021) on the distribution of secondary metabolite clusters in B. gladioli should be compared and discussed.
Response 7: We added the references to Jones et al. and Lin et al and modified the text to ‘In this regard, Burkholderia-produced NRP and NPRS BGCs have been recently explored in some Burkholderia members, including B. gladioli [32,39]’ (page 15, lines 517–519). Subsequently, we compared our result with the findings of Jones et al on the distribution of secondary metabolite clusters in B. gladioli, and discussed in section 3.5 In silico analysis of nonribosomal peptide synthetases (page 15, lines 524–531).
Compared with the distribution of BGCs in the previously studied B. gladioli by Jones et al., we also consistently identified the distribution of BGCs for bongkrekic and for icosalide in genomes of B. gladioli within specific clades. The BGC for bongkrekic was specifically contained in genome of strain MSMB1756 in clade 1A and in genome of strain BSR3 in clade 1B. The BGC for icosalide was widely distributed in all 14 B. gladioli strains within clades 1A, 1B, 2 and 3 (Figure 2B and 5).
Point 8: Minor points: The Comparison of the average nucleotide identity figure is incorrectly numbered. The number of isolates included in each specific genome analysis should be indicated in figure 4. A rationale for exploring Crispr-Cas systems should be included.
Response 8: We thank the reviewer for this point. We carefully checked and re-numbered figures and tables in the order of their citation in the text.
The information about isolates, which were included in each specific genome analysis, was added following text to the legend of Figure 4: ‘The gladiolus-associated genome was derived from gladiolus isolates (ATCC 10248, KACC 11889, ATCC 25417 and NCTC 12378), the rice- and human-associated genomes were derived from rice isolates (BSR3 and KACC 18962), and human isolates (AU0032, AU26456, AU29541 and AU30473), respectively. The nature-associated genome was derived from nature isolates (Coa14, MSMB1756, FDAARGOS_390 and FDAARGOS_391).’.
We addressed the rationale for exploring CRISPR/Cas system in section 3.6 Prediction of CRISPR/Cas system (page 16, lines 553–561).

Reviewer 2 Report
The paper "Pan-genome Analysis Reveals Host-specific Functional Divergences in Burkholderia gladioli" is well written, each analysis is well determined and worth to be published in Microorganisms.
Author Response
Point 1: The paper "Pan-genome Analysis Reveals Host-specific Functional Divergences in Burkholderia gladioli" is well written, each analysis is well determined and worth to be published in Microorganisms.
Response 1: We appreciate the time and effort that the reviewer dedicated to providing feedback on our manuscript and are grateful for the comment.
Round 2
Reviewer 1 Report
I thank the authors for their response to the initial peer review comments, and the improvements made to the manuscript in light of these comments. My initial comments focused on the size of the data set, and I would ideally like to see a clade-specific analysis from a much larger set of B. gladioli genomes to see how widely the conclusion made in the manuscript can be applied to the species. However, I acknowledge the technical and other restrictions raised by the authors, and agree in the value of this exploratory work on 14 isolates which is an interesting advance in the field, and has a contribution towards establishing how B. gladioli are capable of such diverse niche occupation.
Point 1:
As above, I agree that expansion of the data set to include more genomes may result in reduction in the ‘niche-associated’ genes identified, and that lower quality sequence inclusion is likely to have a problematic effect. For that reason, and with the modifications already made to the manuscript, I am happy that this study gives meaningful analysis of a representative and diverse subset of B. gladioli. It is made very clear that this analysis relates to a 14-genome subset and is therefore appropriate as an exploratory investigation into niche adaptation of B. gladioli.
Point 2:
I am very pleased to see the development of Figure 2B, which allows the 14 isolates from this study to be placed into the wider context of known B. gladioli diversity. The overlap of isolation source groupings from this study and phylogenomic groupings form previous studies can now be directly observed. This is of value to the reader since the degree of diversity covered is clearer. This does however reveal the limited diversity of the Human isolates in this study, which come from a single part of clade 3, while isolates from across the tree are known to be able to infect CF patients. More than other niches, I think the ‘human’ niche is most underrepresented. My burning question is if a broader diversity of human isolated B. gladioli were included, would the DNA repair, recombination and biofilm gene enrichment still be seen? I am happy for this to be addressed by an emphasis in the conclusions around line 484 along the lines that the results obtained reflect just four human isolates from one phylogenomic clade and remains to be seen if this is a conserved feature common to more diverse human isolates of B. gladioli.
Point 3:
See my response to point 1. I am happy that this is addressed and have no further comments related to point 3.
Point 4:
Section 270-282 is a welcome addition which acknowledges the diversity of the subset of isolates. All changes are adequate.
Point 5:
The addition of more specificity for ‘nature’ isolates is a welcome addition. The lack of specific information in the NCBI database is a frustration, but does not affect the manuscript.
Point 6:
The changes to the wording better reflect the outcomes of the work.
May I ask that line 603 make a more specific conclusion: “Our pan-genomic study investigated the relationship between fourteen B. gladioli genomes and the ecological niche of their isolation.”
Point 7:
The section on secondary metabolites is now much improved, and it is clearer where the authors have confirmed the findings of others, and more importantly, where they have made advances on our understanding of secondary metabolite distribution in this species. I would replace ‘under-investigated’ with ‘not investigated’ in line 53.
Point 8:
All parts of point 8 suitably addressed.
With the few minor alternations mentioned above, I believe this manuscript is acceptable for publication.
